# Vertical Orientation of Liquid Crystal on 4-*n*-Alkyloxyphenoxymethyl-Substituted Polystyrene Containing Liquid Crystal Precursor

**DOI:** 10.3390/polym13050736

**Published:** 2021-02-27

**Authors:** Kyutae Seo, Hyo Kang

**Affiliations:** BK-21 Four Graduate Program, Department of Chemical Engineering, Dong-A University, 37 Nakdong-Daero 550 Beon-gil, Saha-gu, Busan 604-714, Korea; kyutae@donga.ac.kr

**Keywords:** anisotropic material, liquid crystal, orientation layer, vertical, 4-*n*-alkyloxyphenol

## Abstract

We synthesized a series of polystyrene derivatives that were modified with precursors of liquid crystal (LC) molecules, such as 4-ethyloxyphenol (homopolymer PEOP and copolymer PEOP#; # = 20, 40, 60, and 80, where # indicates the molar fraction of 4-ethyloxyphenoxymethyl in the side chain), 4-*n*-butyloxyphenol (PBOP), 4-*n*-hexyloxyphenol (PHOP), and 4-*n*-octyloxyphenol (POOP), via polymer modification reaction to investigate the orientation of LC molecules on polymer films, exhibiting part of the LC molecular structure. LC molecules showed a stable and uniform vertical orientation in LC cells fabricated with polymers that have 4-ethyloxyphenoxymethyl in the range of 40–100 mol%. In addition, similar results were obtained in LC cells fabricated with homopolymers of PEOP, PBOP, PHOP, and POOP. The vertical orientation of LC molecules in LC cells fabricated with polymer films correlated to the surface energy of polymer films. For example, vertical LC orientation was observed when the total surface energies of the polymer films were lower than approximately 43.2 mJ/m^2^. Good alignment stabilities were observed at 150 °C and 20 J/cm^2^ of ultraviolet irradiation for LC cells fabricated with PEOP film.

## 1. Introduction

The macroscopic physicochemical properties of materials that are composed of anisotropic molecules are affected by their molecular orientations [1,2,3,4,5,6,7]. Extensive studies have investigated the physicochemical properties of anisotropic materials, including thermal conductivity [8,9,10,11], mechanical properties [12,13,14], wettability [15,16], and ionic conductivity [17,18,19], with respect to molecular orientations. Initially, the thermal conductivity of anisotropic materials could be controlled by adjusting the orientation of the anisotropic molecules in the polymer chain. The polymer chains oriented parallel to the heat transfer direction are preferred for increasing thermal conductivity because strong carbon–carbon covalent bonds transport atomic vibrational energy compared to weak van der Waals interchain interactions [8,9]. For example, the thermal conductivity of single-crystal polyethylene in parallel to the polymer chain is considerably greater than that perpendicular to the polymer chain, owing to the aforementioned mechanism [20]. The mechanical properties of a material are closely related to the orientation of anisotropic molecules. The tensile strength of materials with anisotropic molecules parallelly oriented to the applied load is greater than that with anisotropic molecules perpendicularly oriented to the load. For example, thermoplastics reinforced with glass fiber exhibit impressive mechanical properties according to the glass fiber orientation. The maximum tensile strength was observed for glass fibers oriented parallelly to the external stress [12,13,14]. In addition, the surface property of polymeric materials could be reformed by the introduction of anisotropic molecules onto the surface. For example, polydimethylsiloxane (PDMS) is restricted to applications in biomedical devices because of the fast hydrophobic recovery in vivo and/or in vitro system [21,22,23,24], despite the numerous advantages, such as low cost, ease of fabrication, and optical transparency [25,26,27,28,29]. The surface properties of PDMS for biomedical applications, such as biosensors, bioanalytical devices, and implants, could be changed by orienting the anisotropic molecules, such as the amphiphilic surfactant, vertically onto the polymer surface. The hydrophobic tails of the anisotropic surfactants are adsorbed onto the PDMS surface while the hydrophilic head sticks out into the aqueous solution, thereby changing the PDMS surface properties [15,16]. The control of molecular scale ordering via self-assembly in nanotechnology is indispensable, owing to its necessity in developing high-performance materials. Liquid crystal (LC) molecules, which are anisotropic materials, have numerous applications, owing to unique characteristics, such as dynamic molecular order, self-assembling, and anisotropic optical, electrical, and magnetic properties. For example, ionic LC molecules are exceptional candidates for efficient ion conduction because LCs form well-organized channels for ion transportation in their LC phases. The anisotropic properties and processability of LCs have high potentiality in advanced areas, such as biosensors and drug delivery [17,18,19]. It is important to orient the anisotropic molecules, such as LC molecules, on the substrate in one direction, as this plays an essential role in the LC orientation technique for diverse applications [30], as described earlier. In general, the rubbing technique is a method to make uniformly oriented anisotropic molecules [31]. The rubbing of polymeric surfaces is the most common technique to produce uniform orientation of LC molecules in the fabrication of electro-optical devices [32,33,34,35,36]. Through a contact method such as the rubbing technique, aromatic polymers with a rigid backbone, such as polyimide derivatives, are commonly employed as LC orientation substrates because they provide very stable LC orientation via strong interactions, such as *π*–*π* and dipole–dipole interactions, between polymer and LC molecules [37,38,39,40,41,42,43]. Moreover, polyimide derivatives having long alkyl or alkyloxy groups show vertical LC orientation behavior [44,45,46,47]. However, the vertical LC orientation layer on polystyrene (PS) derivatives with long alkyl or fluoroalkyl groups can be synthesized using non-contact methods because long alkyl or fluoroalkyl groups on polystyrene layers produce low surface energy, owing to the steric effect of alkyl or fluoroalkyl groups on the polymer film surface [48]. The surface energy of polymer films and the molecular orientation in polymers are decisive factors in obtaining vertical LC orientation behaviors, owing to different steric repulsions and/or interactions between LC molecules and surfaces.

In this study, we synthesized a series of polystyrene derivatives that have 4-*n*-alkyloxyphenoxymethyl side groups (Figure 1) to systematically investigate the LC orientation behavior of the polystyrene derivates. The synthesis and characterization of these polymers and the optical properties of assembled LC cells with unrubbed polymer films were studied.

## 2. Materials and Methods

### 2.1. Materials

The 4-chloromethylstyrene and 4′-pentyl-4-biphenylcarbonitrile (5CB, *n_e_* = 1.7360, *n_o_* = 1.5442, and Δ*ε* = 14.5, where *n_e_*, *n_o_*, and Δ*ε* represent the extraordinary refractive indexes, ordinary refractive indexes, and dielectric anisotropy, respectively) were purchased from Merck Co. The 4-ethyloxyphenol, 4-*n*-butyloxyphenol, 4-*n*-hexyloxyphenol, and 4-*n*-octyloxyphenol were obtained from Tokyo Chemical Industry (TCI) Co. (Tokyo, Japan). The potassium carbonate, 2,2′-azoisobutyronitrile (AIBN), tetrahydrofuran (THF), and *N,N*′-dimethylacetamide (DMAc) were acquired from Daejung Co. (Busan, Korea). The methanol was supplied by SK Chemical Co. (Ulsan, Korea). The DMAc and ethanol were dried over molecular sieves (4 Å). The THF was dried through refluxing with benzophenone and sodium, followed by distillation. The 4-chloromethylstyrene was purified using column chromatography on silica gel, with hexane as the eluent to remove impurities and inhibitors (*tert*-butylcatechol and nitroparaffin). The AIBN was purified through crystallization using methanol. Poly(4-chloromethylstyrene) (PCMS) was synthesized through the conventional free radical polymerization of 4-chloromethylstyrene using AIBN under a nitrogen atmosphere. The solution mixture was cooled to room temperature and then poured into methanol to obtain a white precipitate. The precipitate was further purified through Soxhlet extraction using hot methanol to remove the remaining monomer (4-chloromethylstyrene) and low molecular weight PCMS. The AIBN was used as an initiator. Other reagents and solvents were used as received.

^1^H NMR of PCMS (400 MHz, CDCl_3_, *δ*/ppm): *δ* = 1.01–1.88 (–*CH_2_*–*CH*–Ph–, 3H), *δ* = 4.13–4.77 (–Ph–*CH_2_*–Cl, 2H), *δ* = 6.00–7.22 (CH_2_–CH–*PhH*–CH_2_–, 4H).

### 2.2. Preparations of 4-n-Alkyloxyphenoxymethyl Modified Polystyrene

The following procedure was used to synthesize 4-*n*-alkyloxyphenoxymethyl-substituted polystyrenes (PAOPs), where the alkyl group is –O–(CH_2_)_n_H (*n* = 2, 4, 6, and 8). The 4-ethyloxyphenoxymethyl-substituted polystyrene (PEOP) synthesis is given as an example. A mixture of PCMS (0.300 g, 1.97 mmol), 4-ethyloxyphenol (0.407 g, 2.96 mmol, 150 mol% compared to PCMS), and potassium carbonate (0.489 g, 3.54 mmol, 120 mol% compared to 4-ethyloxyphenol, used as a substituent) in DMAc (50 mL) was heated to 70 °C and magnetically stirred at 200 rpm under nitrogen atmosphere for 24 h. Thereafter, the solution mixture was cooled to room temperature and then poured into methanol to obtain a white precipitate. The precipitate was further purified by several reprecipitations from the DMAc solution into methanol, and Soxhlet extractor was used to remove the potassium carbonate and remaining salts with hot methanol. A yield of 80% PEOP was obtained after overnight drying under vacuum conditions.

^1^H NMR of PEOP (400 MHz, CDCl_3_, *δ*/ppm): *δ* = 0.98–2.38 (*–CH_2_–CH–*Ph–CH_2_–, –O–CH_2_*–CH_3_,* 6H), *δ* = 3.69–4.04 (–Ph–O*–CH_2_–*CH_3_, 2H), *δ* = 4.56–5.02 (–Ph*–CH_2_–*O–Ph–O–, 2H), *δ* = 6.20–7.21 (–CH_2_–CH–*PhH*–CH_2_–O–, –CH_2_–O–*PhH*–O–, 8H).

Similarly, 4-*n*-butyloxyphenoxymethyl (PBOP, *n* = 4), 4-*n*-hexyloxyphenoxymethyl (PHOP, *n* = 6), and 4-*n*-octyloxyphenoxymethyl-substituted polystyrene (POOP, *n* = 8) were synthesized using the aforementioned procedure. Here, 4-*n*-butyloxyphenol (0.490 g, 2.95 mmol, 150 mol% compared with PCMS), 4-*n*-hexyloxyphenol (0.573 g, 2.95 mmol, 150 mol% compared with PCMS), and 4-*n*-octyloxyphenol (0.656 g, 2.95 mmol, 150 mol% compared with PCMS), respectively, replaced 4-ethyloxyphenol.

^1^H NMR of PBOP (400 MHz, CDCl_3_, *δ*/ppm): *δ* = 0.80–1.82 (*–CH_2_–CH–*Ph–CH_2_–, –O–CH_2_–(*CH_2_*)_2_*–CH_3_*, 10H), *δ* = 3.69–3.95 (–Ph–O*–CH_2_–*(CH_2_)_2_–CH_3_, 2H), *δ* = 4.64–4.97 (–Ph*–CH_2_–*O–Ph–O–, 2H), *δ* = 6.20–7.21 (–CH_2_–CH–*PhH*–CH_2_–O–, –CH_2_–O–*PhH*–O–, 8H).

^1^H NMR of PHOP (400 MHz, CDCl_3_, *δ*/ppm): *δ* = 0.80–1.82 (*–CH_2_–CH–*Ph–CH_2_–, –O–CH_2_–(*CH_2_*)_4_*–CH_3_,* 14H), *δ* = 3.72–3.93 (–Ph–O*–CH_2_–*(CH_2_)_4_–CH_3_, 2H), *δ* = 4.60–4.96 (–Ph*–CH_2_–*O–Ph–O–, 2H), *δ* = 6.20–7.21 (–CH_2_–CH–*PhH*–CH_2_–O–, –CH_2_–O–*PhH*–O–, 8H).

^1^H NMR of POOP (400 MHz, CDCl_3_, *δ*/ppm): *δ* = 0.80–1.80 (*–CH_2_–CH–*Ph–CH_2_–, –O–CH_2_–(*CH_2_*)_6_*–CH_3_,* 18H), *δ* = 3.73–3.97 (–Ph–O*–CH_2_–*(CH_2_)_6_–CH_3_, 2H), *δ* = 4.62–5.00 (–Ph*–CH_2_–*O–Ph–O–, 2H), *δ* = 6.20–7.21 (–CH_2_–CH–*PhH*–CH_2_–O–, –CH_2_–O–*PhH*–O–, 8H).

The copolymer of PEOP, designated as PEOP#, where # is the degree (mol%) of substitution of chloromethyl to the 4-ethyloxyphenoxymethyl group, were prepared using the same procedure as for the PEOP. However, less than 150 mol% of 4-ethyloxyphenol was used. For example, PEOP20, PEOP40, PEOP60, and PEOP80 were prepared with 4-ethyloxyphenol of 0.054 g (0.39 mmol), 0.109 g (0.79 mmol), 0.163 g (1.18 mmol), and 0.217 g (1.57 mmol), respectively, using a slight excess of potassium carbonate (120 mol% compared to 4-ethyloxyphenol).

^1^H NMR of PEOP20 (400 MHz, CDCl_3_, *δ*/ppm): *δ* = 0.97–2.38 (*–CH_2_–CH–*Ph–CH_2_–Cl, *–CH_2_–CH–*Ph–CH_2_–O–, –O–CH_2_*–CH_3_,* 9H), *δ* = 3.69–4.04 (–Ph–O*–CH_2_–*CH_3_, 2H), *δ* = 4.56–5.02 (–Ph*–CH_2_–*Cl, 2H) *δ* = 4.56–5.02 (–Ph*–CH_2_–*O–Ph–O–, 2H), *δ* = 6.20–7.21 (–CH_2_–CH–*PhH*–CH_2_–Cl, –CH_2_–CH–*PhH*–CH_2_–O–, –CH_2_–O–*PhH*–O–, 12H).

^1^H NMR of PEOP40 (400 MHz, CDCl_3_, *δ*/ppm): *δ* = 0.97–2.38 (*–CH_2_–CH–*Ph–CH_2_–Cl, *–CH_2_–CH–*Ph–CH_2_–O–, –O–CH_2_*–CH_3_,* 9H), *δ* = 3.69–4.04 (–Ph–O*–CH_2_–*CH_3_, 2H), *δ* = 4.56–5.02 (–Ph*–CH_2_–*Cl, 2H) *δ* = 4.56–5.02 (–Ph*–CH_2_–*O–Ph–O–, 2H), *δ* = 6.20–7.21 (–CH_2_–CH–*PhH*–CH_2_–Cl, –CH_2_–CH–*PhH*–CH_2_–O–, –CH_2_–O–*PhH*–O–, 12H).

^1^H NMR of PEOP60 (400 MHz, CDCl_3_, *δ*/ppm): *δ* = 0.97–2.38 (*–CH_2_–CH–*Ph–CH_2_–Cl, *–CH_2_–CH–*Ph–CH_2_–O–, –O–CH_2_*–CH_3_,* 9H), *δ* = 3.69–4.04 (–Ph–O*–CH_2_–*CH_3_, 2H), *δ* = 4.56–5.02 (–Ph*–CH_2_–*Cl, 2H) *δ* = 4.56–5.02 (–Ph*–CH_2_–*O–Ph–O–, 2H), *δ* = 6.20–7.21 (–CH_2_–CH–*PhH*–CH_2_–Cl, –CH_2_–CH–*PhH*–CH_2_–O–, –CH_2_–O–*PhH*–O–, 12H).

^1^H NMR of PEOP80 (400 MHz, CDCl_3_, *δ*/ppm): *δ* = 0.97–2.38 (*–CH_2_–CH–*Ph–CH_2_–Cl, *–CH_2_–CH–*Ph–CH_2_–O–, –O–CH_2_*–CH_3_,* 9H), *δ* = 3.69–4.04 (–Ph–O*–CH_2_–*CH_3_, 2H), *δ* = 4.56–5.02 (–Ph*–CH_2_–*Cl, 2H) *δ* = 4.56–5.02 (–Ph*–CH_2_–*O–Ph–O–, 2H), *δ* = 6.20–7.21 (–CH_2_–CH–*PhH*–CH_2_–Cl, –CH_2_–CH–*PhH*–CH_2_–O–, –CH_2_–O–*PhH*–O–, 12H).

### 2.3. Film Preparation and LC Cell Assembly

Solutions of PEOP#, PAOP (PEOP, PBOP, PHOP, and POOP) in THF (1 wt.%) were filtered using a poly(tetrafluoroethylene) (PTFE) membrane with a pore size of 0.45 μm. Then, thin polymer films were prepared by spin-coating (2000 rpm, 60 s) onto 2.0 × 2.5 cm^2^ glass substrates. LC cells were fabricated by assembling two polymeric layers onto the two glass substrates using spacers with a thickness of 4.25 μm. The physicochemical properties, such as surface tension, of the 4′-pentyl-4-cyanobiphenyl (5CB) have been documented in numerous studies because of its accessible nematic temperature range around room temperature, high positive dielectric anisotropy, and remarkable chemical stability. Therefore, the 5CB was selected to fabricate LC cells in order to investigate the correlation between the orientation layer and LC molecules via physicochemical interaction [49,50,51,52]. The cells were filled with nematic LC, 5CB. The manufactured LC cells were sealed with epoxy glue.

### 2.4. Instrumentation

For characterization of the synthesized structure, ^1^H nuclear magnetic resonance (NMR) spectroscopy, using an Agilent MR400 DD2 NMR spectrometer, differential scanning calorimetry (DSC) (TA Instruments, Q-10), and polarized optical microscopy (POM) images of the LC cells, using a Nikon Eclipse E600 POL, NIKON Co. (Tokyo, Japan), installed polarizer, and digital camera (Nikon, Coolpix 995, NIKON Co., Tokyo, Japan), were performed. The energy dispersive spectroscopy (EDS) mapping analysis was performed using dual-beam focused ion beam (FIB) scanning electron microscopy (SEM) fitted with an Oxford EDS (Thermo Fisher Scientific, Scios2) in order to confirm the uniformity and the thermal stability of the polymer film on the glass substrate. The contact angles of distilled water and methylene iodide on PAOP and PEOP# films were determined using a Kruss DSA10 contact angle analyzer equipped with drop shape analysis software. The surface energy was measured using the Owens–Wendt equation, given below:γ*_sl_*_=_ γ*_s_* + γ*_l_* − 2(γ*_s_^d^*γ*_l_^d^*)^1/2^ − 2(γ*_s_^p^*γ*_l_^p^*)^1/2^(1)
where γ*_sl_* is the surface energy of the solid–liquid interface, γ*_s_* is the surface energy of the solid, γ*_s_^d^* is the dispersive component of surface energy, γ*_s_^p^* is the polar component of surface energy, γ*_l_* is the liquid surface tension, γ*_l_^d^* is the dispersive component of surface tension, and γ*_l_^p^* is the polar component of surface tension. The surface energy (γ*_s_*) of the solid is the sum of dispersive (γ*_s_^d^*) and polar (γ*_s_^p^*) components of surface energy [53].

## 3. Results and Discussion

The synthesis routes for PAOPs (PEOP, PBOP, PHOP, and POOP) and PEOP# are shown in Figure 1. The copolymers that have different substitution ratios (mol%) were obtained by varying the molar ratio of 4-ethyloxyphenol in the reaction mixture. An approximately complete conversion of chloromethyl to oxymethyl was obtained when an excess (150 mol%) of 4-ethyloxyphenol, 4-*n*-butyloxyphenol, 4-*n*-hexyloxyphenol, and 4-*n*-octyloxyphenol, respectively, were reacted with poly(4-chloromethylstyrene) at 70 °C for 24 h. Figure 2 shows the ^1^H NMR spectrum of PEOP as an example. Chemical compositions of the monomeric units in the obtained polymers were confirmed using the ^1^H NMR spectrum. The ^1^H NMR spectrum and the assignment of the respective peaks of PEOP are shown in Figure 2. The ^1^H NMR spectrum of PEOP indicates the presence of protons from PEOP derivatives ((*δ*/ppm = 0.98–2.38 (3H), 4.56–5.02 (2H), 3.69–4.04 (2H), and 6.20–7.21 (4H)); peaks a, b, c, and d). The degree of substitution from chloromethyl to oxymethyl was calculated to be approximately 100% by comparing the integration area of the oxymethyl peak at 4.56–5.02 ppm and the backbone peaks at 0.98–2.38 ppm. Similar integrations and calculations for PEOP# and PAOP were performed and were typically within ±10% of the expected values. The ^1^H NMR spectra of the other polymers, including PAOPs and PEOP#, are shown in Appendix A. These polymers were soluble in medium-polarity solvents with low boiling points, such as tetrahydrofuran and chloroform, and in aprotic polar solvents, such as *N,N*′-dimethylformamide (DMF) and *N,N*′-dimethylacetamide (DMAc). The solubility of polymer samples in various solvents was sufficient for PEOP# and PAOP to be applied as thin film materials. Among the organic solvents, THF was chosen as the coating solvent for thin film fabrication, owing to its low eco-toxicity and good biodegradability [54]. In addition, these polymer thin films could be fabricated using a low-temperature process based on a wet process, owing to good solubility in volatile organic solvents. The energy dispersive spectroscopy (EDS) mapping images of the bare glass and the polymer films on the glass substrates before and after heat treatment at 200 °C for 10 min were observed at different positions, in order to determine coating failure and thermal stability of the polymer films. For example, the coating uniformity of the PEOP film on the glass substrate using THF was confirmed by the carbon mapping images in Appendix A. The discernible difference in the number of carbon elements in the PEOP film on the glass substrate before and after heat treatment could not be observed by the EDS mapping images, as illustrated in Appendix A. Therefore, it could be interpreted that the polymer film on the glass substrate has satisfactory uniformity and thermal stability.

The polymer thermal properties were studied using differential scanning calorimetry (DSC) at a heating and cooling rate of 10 °C/min under a nitrogen atmosphere. All polymers were amorphous; only one glass transition was observed in their DSC thermograms. The glass transition temperatures were determined from the extrapolated intersection of the asymptotes to the glassy and rubbery regions for the enthalpy [55,56], as illustrated in Figure 3. As the molar content of 4-ethyloxyphenoxymethyl side group increased from 20 mol% to 100 mol%, the *T_g_* value decreased from 100 °C for PEOP20 to 61.2 °C for PEOP (Table 1). In addition, as the number of carbon atoms in the alkyl moiety of the 4-alkyloxypheoxymethyl side group increased from 2 to 6, *T_g_* decreased from 61.2 °C for PEOP to 37.7 °C for PHOP. The decrease of polystyrene derivative *T_g_*s with increasing bulkiness of the side groups was reported earlier and ascribed to the increase of free volume in the polymer because polymers that have larger free volumes have lower *T_g_*s [57,58,59]. However, as the number of carbon atoms of the side groups increased from 6 to 8, *T_g_* increased from 37.7 °C for PHOP to 38.4 °C for POOP. The increase of polystyrene derivative *T_g_*s was ascribed to the increase of the interactions of side groups, such as *π*–*π* and van der Waals interactions [57,58,60,61].

It has been shown that the LC molecular orientations could be affected by the chemical structure of the orientation layer because of the interaction at the interface between LC molecules and the orientation layer [62,63,64]. We fabricated the LC cells made from films of polystyrene derivatives grafted with precursors of LC molecules, such as 4-ethyloxyphenoxymethyl, 4-*n*-butyloxyphenoxymethyl, 4-*n*-hexyloxyphenoxymethyl, and 4-*n*-octyloxyphenoxymethyl, using 5CB to investigate the orientation behavior of LC molecules on the polymer films that have LC-like moieties. Figure 4a shows the photograph of the LC cells fabricated with the PAOP films (PEOP, PBOP, PHOP, and POOP). These LC cells showed uniform vertical LC orientation behavior in the whole area under two crossed polarizers. The polarized optical microscopy (POM) images of the LC cells were obtained in orthoscopic (top) and conoscopic modes (bottom) in Figure 4(b) to systematically investigate the orientation of the LC molecules. The vertical orientation of the LC cells is shown in a Maltese cross pattern in the conoscopic POM images. In addition, the LC cells assembled with PAOP films had orientation stability over several months. The vertical orientation of the LC molecules might be explained by the similarity of molecular structures between the orientation layer and the LC molecules.

In addition, we observed the orientation behavior of the LC molecules in PEOP# cells, using 5CB to investigate the effect of the molar content of the 4-ethyloxyphenoxymethyl side group in the polymer. Figure 5a shows the photographs of the LC cells made from PEOP# and PEOP. The PEOP20 LC cells exhibited LC textures with birefringence, while good uniformity of vertical LC orientation was observed for PEOP40, PEOP60, PEOP80, and PEOP in the whole area. Additionally, LC orientation behaviors of the LC cells fabricated with the PEOP# films were investigated by observing their POM images for close examination in Figure 5b. When the molar fraction of the 4-ethyloxyphenoxymethyl containing a monomeric unit in the PEOP# was less than 20 mol%, the LC cells prepared from PEOP20 film showed random planar LC orientation with birefringence in the conoscopic POM image. In contrast, PEOP40, PEOP60, PEOP80, and PEOP provided stable vertically oriented LC layers. The discernible difference in LC orientation on PEOP40, PEOP60, PEOP80, and PEOP films, according to the molar fraction of the 4-ethyloxyphenoxymethyl in the side groups, could not be observed using the Maltese cross pattern in the conoscopic POM images. The vertical alignment of the LC cells made of polymer films with short and small molar contents of side groups and, long and high molar content of side groups was observed. We believe that these results show that the similarity of the molecular structure between the LC molecules and the alignment layer helps orient the LC molecules vertically to the polymeric surface.

We investigated the LC orientation behaviors of PEOP# and PAOP films using surface characterization techniques, such as surface energy measurement, based on the static contact angles of water and methylene iodide, and they are shown in Figure 6 and Table 2, respectively. The surface energies were calculated using the Owens–Wendt equation. The total surface energies of the respective polymer films were calculated from the summation of polar and dispersion components. It has been widely known that the orientation of LC molecules on the orientation layer could be explained by the total surface energy of the orientation layer. For example, the LC molecules have a tendency to be oriented vertically onto the orientation film in order to maximize their intermolecular interaction when the total surface energy of the orientation film is relatively low [65,66,67]. The vertical LC orientation of PEOP40, PEOP60, PEOP80, PEOP, PBOP, PHOP, and POOP was observed. The total surface energies of these polymers are 43.2, 43.2, 42.9, 42.4, 39.6, 32.7, and 21.0 mJ/m^2^, respectively. However, PEOP20 (46.7 mJ/m^2^) did not show vertical LC orientation behavior. Therefore, the vertical LC orientation behavior correlates well with the total surface energy of the polymer, having less than approximately 46.7 mJ/m^2^ (the critical surface energy of the polymer films to induce vertical LC orientation) [68,69].

The reliability of the LC cells fabricated with polymer films was substantiated via the evaluation of the LC aligning stability under harsh conditions, such as high temperatures and ultraviolet (UV) irradiation. Figure 7a shows the thermal stabilities of the LC cells fabricated with PEOP film, estimated using the POM images after heating the LC cells for 10 min at temperatures of 100, 150, and 200 °C, respectively. The POM images of the LC cells fabricated with polymer film indicate that the vertical LC alignment was maintained for 10 min at 150 °C. Therefore, the processing temperature of this polymer for LC cell applications must be below 150 °C. In addition, the UV stabilities of the LC cells fabricated with PEOP film were estimated from conoscopic POM images. The conoscopic POM images of the LC cells were captured after UV irradiation at 5, 10, and 20 J/cm^2^, respectively. As shown in Figure 7b, discernible differences in the vertical LC orientation on PEOP film were not observed in the conoscopic POM images, indicating that the vertical LC alignment of the LC cells was maintained even at high UV irradiations.

## 4. Conclusions

A series of 4-ethyloxyphenoxymethyl-substituted polystyrenes (PEOP# and PEOP), PBOP, PHOP, and POOP-substituted polystyrenes were synthesized to evaluate the LC orientation behaviors of their polymer films. The vertical LC orientation behavior was observed for the LC cells from polymers that have a higher molar content of 4-ethyloxyphenoxymethyl side groups. For example, LC cells with greater than 40 mol% of 4-ethyloxyphenoxymethyl (PEOP40, PEOP60, PEOP80, and PEOP) showed vertical LC orientation, while LC cells fabricated with PEOP20 films, having less than a 20 mol% of the 4-ethyloxyphenoxymethyl group, exhibited a random planar LC orientation. The vertical orientation of LC molecules in LC cells fabricated with polymer films was observed, despite the short side chain length (PEOP) and low substitution ratio (about 40 mol%). Moreover, LC precursor structures in the polymer side chains helped orient vertical LC orientations through *π*–*π* and van der Waals interactions between polymer chains and LC molecules. The vertical LC orientation behavior correlated well with polymer films that have total surface energies less than approximately 46.70 mJ/m^2^, owing to the unique structure of the 4-*n*-alkyloxyphenoxymethyl side chain. Therefore, 4-*n*-alkyloxyphenoxymethyl-substituted polystyrenes are a potential candidate for LC orientation layers, with next-generation applications with low-temperature wet processes.

## Figures and Tables

**Figure 1 polymers-13-00736-f001:**
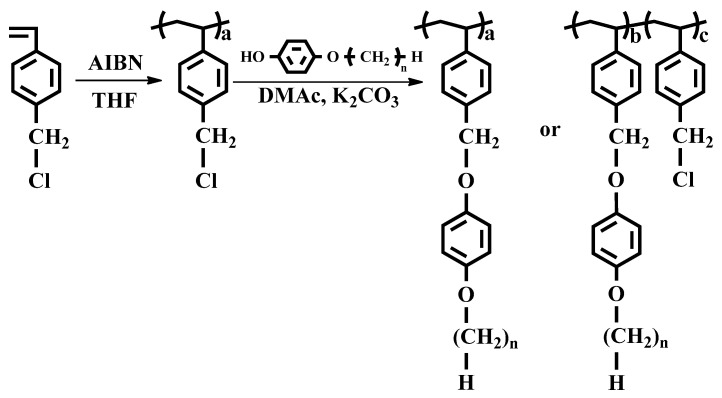
Synthetic route to 4-ethyloxyphenoxymethyl (PEOP# and PEOP, *n* = 2), 4-*n*-buthyloxyphenoxymethyl (PBOP, *n* = 4), 4-*n*-hexyloxyphenoxymethyl (PHOP, *n* = 6), and 4-*n*-octyloxyphenoxymethyl-substituted polystyrene (POOP, *n* = 8), where # represents the molar fraction of 4-ethyloxyphenoxymethyl containing monomeric units in the polymer.

**Figure 2 polymers-13-00736-f002:**
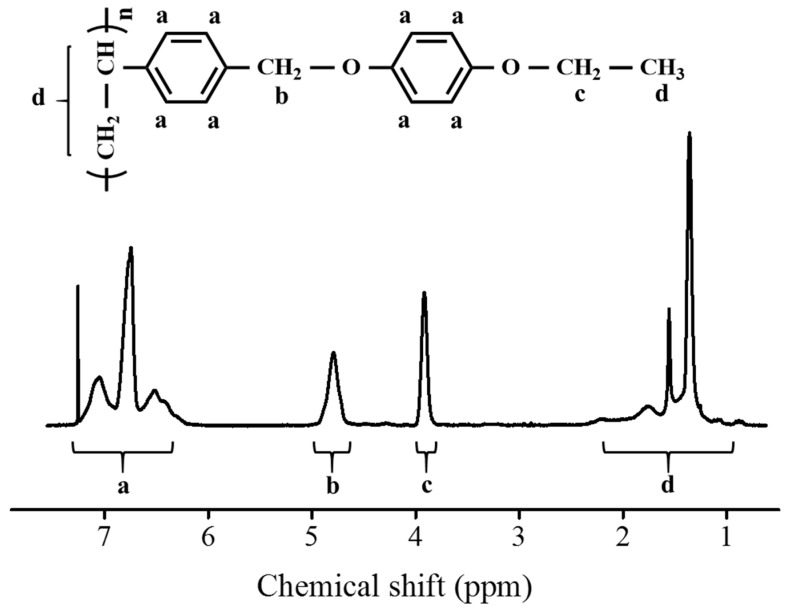
^1^H nuclear magnetic resonance (NMR) spectrum of PEOP.

**Figure 3 polymers-13-00736-f003:**
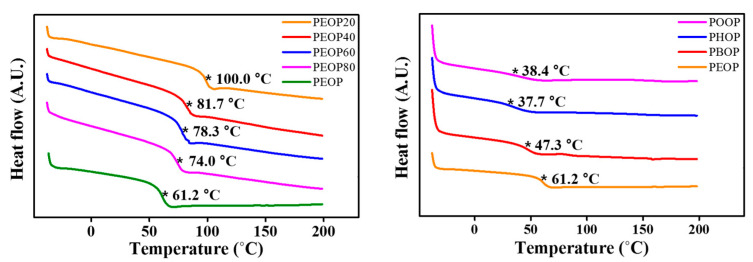
Differential scanning calorimetry (DSC) thermogram of PEOP# (PEOP20, PEOP40, PEOP60, and PEOP80) and PAOP (PEOP, PBOP, PHOP, and POOP). (* means the status of glass transition).

**Figure 4 polymers-13-00736-f004:**
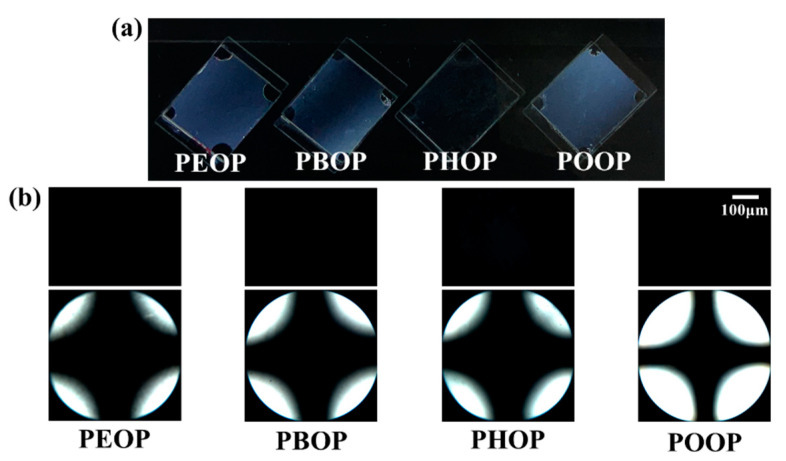
(**a**) Photograph, (**b**) orthoscopic (top) and conoscopic (bottom) polarized optical microscopy (POM) images of the LC cells fabricated with PEOP, PBOP, PHOP, and POOP films.

**Figure 5 polymers-13-00736-f005:**
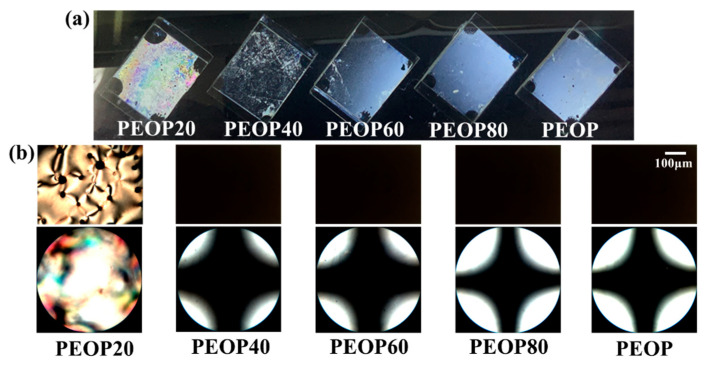
(**a**) Photograph, (**b**) orthoscopic (top) and conoscopic (bottom) polarized optical microscopy (POM) images of the LC cells fabricated with PEOP# (PEOP20, PEOP40, PEOP60, PEOP80, and PEOP).

**Figure 6 polymers-13-00736-f006:**
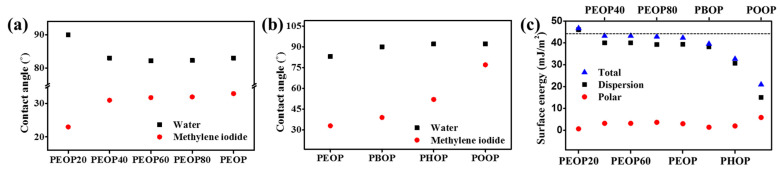
Contact angle of water and methylene iodide on polymer films fabricated with (**a**) PEOP# and (**b**) PAOP, (**c**) surface energy and LC orientation behaviors. Upper and lower parts around the broken line indicates random planar and vertical LC orientation behaviors, respectively.

**Figure 7 polymers-13-00736-f007:**
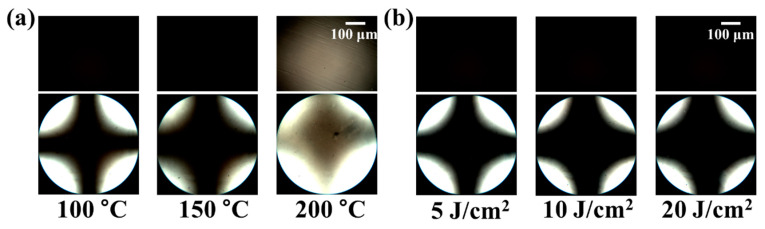
Orthoscopic (top) and conoscopic (bottom) polarized optical microscopy (POM) images of the LC cells made using PEOP films, (**a**) after thermal treatment at 100, 150, and 200 °C for 10 min and (**b**) UV treatment at 5, 10, and 20 J/cm^2^, respectively.

**Table 1 polymers-13-00736-t001:** Reaction conditions and results for the synthesis of the PEOP# and PAOP (PEOP, PBOP, PHOP, and POOP).

Polymer Designation	Feed Ratio of 4-*n*-Alkyloxyphenol (mol%)	Degree of Substitution (mol%)	*T_g_* (°C)
PEOP20	20	20	100.0
PEOP40	40	40	81.7
PEOP60	60	60	78.3
PEOP80	80	80	74.0
PEOP	150	100	61.2
PBOP	150	100	47.3
PHOP	150	100	37.7
POOP	150	100	38.4

**Table 2 polymers-13-00736-t002:** Surface energy and LC orientation properties of the polymers.

Polymer Designation	Contact Angle (°) *^a^*		Surface Energy (mJ/m^2^) *^b^*			Vertical LC Aligning Ability
	Water	Methylene Iodide	Dispersion	Polar	Total	
PEOP20	90	23	46.1	0.6	46.7	No
PEOP40	82	31	40.0	3.2	43.2	Yes
PEOP60	82	31	40.0	3.2	43.2	Yes
PEOP80	82	32	39.3	3.6	42.9	Yes
PEOP	83	33	39.4	3.0	42.4	Yes
PBOP	90	39	38.2	1.4	39.6	Yes
PHOP	92	52	30.7	2.0	32.7	Yes
POOP	92	77	15.1	5.9	21.0	Yes

*^a^* Measured from static contact angle. *^b^* Calculated from the Owens–Wendt equation.

## Data Availability

The data presented in this study are available on request from the corresponding author.

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
