# Peer review of "Vertical Orientation of Liquid Crystal on 4-n-Alkyloxyphenoxymethyl-Substituted Polystyrene Containing Liquid Crystal Precursor"

_polymers, 2021, doi:10.3390/polym13050736_

Round 1
Reviewer 1 Report
1) The authors should better explain the choice of 5CB as LC
2) Are the glass substrates homogeneously covered by the polymer films, how can the authors be sure that no dewetting occurs at high temperatures?
3) The authors should provide a more detailed description of DSC meausurements and data analysis:
- What is the rate used during the heating runs?
- The authors should explain the criterion to determine the glass transition temperature and provide a reference behind
- What is the uncertainty in the Tg values?
4) The symbols of Figure 6 are way to small and makes this figure difficult to read
5) The correlation between the surface energy threshold value and the LC vertical orientation is not clear. It should be better explained in terms of the physichochemical properties of the polymers and LC
Author Response
Dear Editor at Polymers
We gratefully appreciate your kind reviewing and considering for publication in “Polymers”. We are submitting a revised manuscript (polymers-1122863) entitled “Vertical orientation of liquid crystal on 4-n-alkyloxyphenoxymethyl-substituted polystyrene containing liquid crystal precursor”.
We carefully read the reviewer’s comments and your e-mail. Reviewers gave us helpful comments for our manuscript. We think the reviewer’s opinion and suggestion is fairly reasonable. Therefore, we revised our manuscript taking the reviewer’s comments into consideration as follows. As you and the Reviewer suggested we modified some parts of the manuscript and the changes are shown as yellow texts. These changes are listed as follows:
Referee’s comments:
Referee: 1
Comments:
- The authors should better explain the choice of 5CB as LC
Answer:
We deeply thanks for comment suggested by Reviewer and we have added the explanation about the choice of 5CB as a LC material and method section (page 4, lines 161–171) as “The physicochemical properties such as surface tension of the 4’-pentyl-4-cyanobiphenyl (5CB) have been documented in numerous studies because of its accessible nematic temperature range around room temperature, high positive dielectric anisotropy, and remarkable chemical stability. Therefore, the 5CB was selected to fabricated LC cells in order to investigate the correlation between orientation layer and LC molecules by physicochemical interaction.”. Added literatures are below.
- Hanemann, T.; Haase, W. Crystal structure of 4’-pentyl-4-cyanobiphenyl (5CB). Liq. Cryst. 1995, 19, 699–702.
- Bogi, A.; Faetti, S. Elastic, dielectric and optical constants of 4’-pentyl-4-cyanobiphenyl. Liq. Cryst. 2001, 28, 729–739.
- Maze, C. Determination of nematic liquid crystal elastic and dielectric properties from the shape of a capacitance-voltage curve. Mol. Cryst. Liq. Cryst. 1978, 48, 273–287.
- Schell, K.T.; Porter, R.S. Dielectric studies of highly polar nematic liquid crystals and their mixtures. Liq. Cryst. 1990, 188, 97–103.
- Are the glass substrates homogeneously covered by the polymer films, how can the authors be sure that no dewetting occurs at high temperatures?
Answer:
We deeply thanks for comments suggested by Reviewer and we performed the energy dispersive spectroscopy (EDS) mapping analysis using dual-beam focused ion beam (FIB) scanning electron microscopy (SEM) fitted with an Oxford EDS (Thermo Fisher Scientific, Scios2) in order to confirm the coating uniformity and the thermal stability of the polymer film on the glass substrate and have added the sentences in result and discussion section (page 5, lines 219–228) as “The energy dispersive spectroscopy (EDS) mapping images of the bare glass and the polymer films on the glass substrates before and after heat treatment at 200 °C for 10 min were observed at different positions in order to determine coating failure and thermal stability of the polymer films. For example, the coating uniformity of PEOP film on the glass substrate using THF was confirmed by the carbon mapping images in Figure S8. The discernible difference in amount of carbon elements in the PEOP film on the glass substrate between before and after heat treatment cannot be observed by EDS mapping images, as illustrated in Figure S9. Therefore, it could be interpreted that the polymer film on the glass substrate has satisfactory uniformity and thermal stability.” The figures S8 and S9 have added in supplementary materials as below. We strongly believe that Reviewer will satisfy the answer about this question.
Figure S8. Energy dispersive spectroscopy (EDS) mapping images of (a) the bare glass and (b)–(f) PEOP film on the glass substrate observed at different positions.
Figure S9. Energy dispersive spectroscopy (EDS) mapping images of (a)–(e) PEOP film on the glass substrate after thermal treatment at 200 °C for 10 min observed at different positions.
- The authors should provide a more detailed description of DSC meausurements and data analysis. What is the rate used during the heating runs? The authors should explain the criterion to determine the glass transition temperature and provide a reference behind. What is the uncertainty in the Tg values?
Answer:
We deeply thanks for comment suggested by Reviewer and we have added detailed description of DSC measurements in results and discussion section (page 6, lines 237–238) as “The polymer thermal properties were studied using differential scanning calorimetry (DSC) at a heating and cooling rate of 10 °C/min under nitrogen atmosphere.” and we have explained the criterion to determine the glass transition temperature in results and discission (page 6, lines 239–241) and the Figure 3 has been rectified as below in order to exhibit the glass transition temperature obviously. Added literatures are below. We strongly believe that Reviewer will satisfy the answer about this question.
Figure 3. Differential scanning calorimetry (DSC) thermogram of PEOP# (PEOP20, PEOP40, PEOP60, and PEOP80) and PAOP (PEOP, PBOP, PHOP, and POOP).
- Royall, P.G.; Craig, D.Q.; Doherty, C. Characterisation of the glass transition of an amorphous drug using modulated DSC. Pharm. Res. 1998, 15, 1117–1121.
- Hutchinson, J. Determination of the glass transition temperature: methods correlation and structural heterogeneity. J. Therm. Anal. 2009, 98, 579–589.
- The symbols of Figure 6 are way to small and makes this figure difficult to read.
Answer:
We deeply thanks for comment suggested by Reviewer and we have increased size of the symbols in the Figure 6 in order to improve readability as below.
Figure 6. Contact angle of water and methylene iodide on polymer films fabricated with (a) PEOP# and (b) PAOP, (c) surface energy and LC orientation behaviors. Upper and lower parts around the broken line indicates random planar and vertical LC orientation behaviors, respectively.
- The correlation between the surface energy threshold value and the LC vertical orientation is not clear. It should be better explained in terms of the physichochemical properties of the polymers and LC.
Answer:
We deeply thanks for comment suggested by Reviewer and we have added the explanation in terms of the physichochemical properties of the polymers and LC in results and discussion section (page 8, lines 307–311) as “It has been widely known that orientation of LC molecules on the orientation layer could be explained by the total surface energy of the orientation layer. For example, the LC molecules have a tendency to be oriented vertical onto orientation film in order to maximize their intermolecular interaction when total surface energy of the orientation film is relatively low.”. Added literatures are below.
- Bouchiat, M.A.; Langevin-Cruchon, D. Molecular order at the free surface of a nematic liquid crystal from light reflectivity measurements. Phys. Lett. A 1971, 34, 331–332.
- Haller, I. Alignment and wetting properties of nematic liquids. Appl. Phys. Lett. 1974, 24, 349–351.
- Shafrin, E.G.; Zisman, W.A. Constitutive relations in the wetting of low energy surfaces and the theory of the retraction method of preparing monolayers1. J. Phys. Chem. 1960, 64, 519–524.
We believe that now we answered all of the comments pointed out by the reviewers. I hope that now this paper is publishable in “Polymers”, one of the top journals in polymer science area.
We also believe that this paper is also suitable for publication in “Polymers” from the following reasons.
- We synthesized a series of polystyrene derivatives that are modified with precursors of liquid crystal (LC) molecules such as 4-ethyloxyphenol (homopolymer PEOP and copolymer PEOP#; # = 20, 40, 60, and 80, where # indicates the molar fraction of 4-ethyloxyphenoxymethyl in the side chain), 4-n-butyloxyphenol (PBOP), 4-n-hexyloxyphenol (PHOP), and 4-n-octyloxyphenol (POOP) via polymer modification reaction.
- We fabricated the LC cells made from films of polystyrene derivatives grafted with precursors of LC molecules such as 4-ethyloxyphenoxymethyl, 4-n-butyloxyphenoxymethyl, 4-n-hexyloxyphenoxymethyl, and 4-n-octyloxyphenoxymethyl using 5CB to investigate the orientation behavior of LC molecules on the polymer films having LC-like moieties.
- The vertical orientation of LC molecules in LC cells fabricated with polymer films was observed, despite the short side chain length (PEOP) and low substitution ratio (about 40 mol%). Moreover, LC precursor structures in the polymer side chains help orient vertical LC orientations through π–π and van der Waals interactions between polymer chains and LC molecules. The vertical LC orientation behavior correlated well with polymer films having the total surface energies lesser than approximately 46.70 mJ/m2, owing to the unique structure of the 4-n-alkyloxyphenoxymethyl side chain.
- In addition, these polymer thin films could be fabricated using low temperature process based on wet process owing to good solubility in volatile organic solvents. Therefore, 4-n-alkyloxyphenoxymethyl-substituted polystyrenes are a potential candidate for LC orientation layers with next generation applications with low temperature wet processes.
In view of these achievements, we believe that our work represents a timely methodological advance and breakthrough in the field of polymer science and thus is appropriate for a journal with the scope and wider readership of “Polymers”.
I hereby certify that this manuscript consists of original, unpublished work which is not under consideration for publication elsewhere.
We are excited to share our manuscript with you and look forward to hearing good news from you soon.
Thank you very much for your time and consideration for the process.
Sincerely (on behalf of all authors),
Prof. Hyo Kang
Associate Professor
Department of Chemical Engineering
Dong-A University
Busan 49315, Republic of Korea
Tel: +82 51 200 7720
Fax: +82 51 200 7728
E-mail [email protected]

Reviewer 2 Report
The manuscript presents a few more surface modifying polymers which allow for the convenient fabrication of LC cells. Although not new as a concept the technique allows for high alignement without the usual rubbing of the modifier layer.
The paper should include the NMR spectra of the different compounds either in form of the measured integrals of the selected regions, or as original spectra in the ESI section.
Author Response
Dear Editor at Polymers
We gratefully appreciate your kind reviewing and considering for publication in “Polymers”. We are submitting a revised manuscript (polymers-1122863) entitled “Vertical orientation of liquid crystal on 4-n-alkyloxyphenoxymethyl-substituted polystyrene containing liquid crystal precursor”.
We carefully read the reviewer’s comments and your e-mail. Reviewers gave us helpful comments for our manuscript. We think the reviewer’s opinion and suggestion is fairly reasonable. Therefore, we revised our manuscript taking the reviewer’s comments into consideration as follows. As you and the Reviewer suggested we modified some parts of the manuscript and the changes are shown as yellow texts. These changes are listed as follows:
Referee’s comments:
Referee: 2
Comment:
- The manuscript presents a few more surface modifying polymers which allow for the convenient fabrication of LC cells. Although not new as a concept the technique allows for high alignment without the usual rubbing of the modifier layer. The paper should include the NMR spectra of the different compounds either in form of the measured integrals of the selected regions, or as original spectra in the ESI section.
Answer:
We deeply thanks for comment suggested by Reviewer and we have added new Figures S1–S7: 1H nuclear magnetic resonance (NMR) spectra of PBOP, PHOP, POOP, PEOP20, PEOP40, PEOP60, and PEOP80 in supplementary materials as below. We strongly believe that Reviewer will satisfy the answer about this question.
Figure S1. 1H nuclear magnetic resonance (NMR) spectrum of PBOP.
Figure S2. 1H nuclear magnetic resonance (NMR) spectrum of PHOP.
Figure S3. 1H nuclear magnetic resonance (NMR) spectrum of POOP.
Figure S4. 1H nuclear magnetic resonance (NMR) spectrum of PEOP20.
Figure S5. 1H nuclear magnetic resonance (NMR) spectrum of PEOP40.
Figure S6. 1H nuclear magnetic resonance (NMR) spectrum of PEOP60.
Figure S7. 1H nuclear magnetic resonance (NMR) spectrum of PEOP80.
We believe that now we answered all of the comments pointed out by the reviewers. I hope that now this paper is publishable in “Polymers”, one of the top journals in polymer science area.
We also believe that this paper is also suitable for publication in “Polymers” from the following reasons.
- We synthesized a series of polystyrene derivatives that are modified with precursors of liquid crystal (LC) molecules such as 4-ethyloxyphenol (homopolymer PEOP and copolymer PEOP#; # = 20, 40, 60, and 80, where # indicates the molar fraction of 4-ethyloxyphenoxymethyl in the side chain), 4-n-butyloxyphenol (PBOP), 4-n-hexyloxyphenol (PHOP), and 4-n-octyloxyphenol (POOP) via polymer modification reaction.
- We fabricated the LC cells made from films of polystyrene derivatives grafted with precursors of LC molecules such as 4-ethyloxyphenoxymethyl, 4-n-butyloxyphenoxymethyl, 4-n-hexyloxyphenoxymethyl, and 4-n-octyloxyphenoxymethyl using 5CB to investigate the orientation behavior of LC molecules on the polymer films having LC-like moieties.
- The vertical orientation of LC molecules in LC cells fabricated with polymer films was observed, despite the short side chain length (PEOP) and low substitution ratio (about 40 mol%). Moreover, LC precursor structures in the polymer side chains help orient vertical LC orientations through π–π and van der Waals interactions between polymer chains and LC molecules. The vertical LC orientation behavior correlated well with polymer films having the total surface energies lesser than approximately 46.70 mJ/m2, owing to the unique structure of the 4-n-alkyloxyphenoxymethyl side chain.
- In addition, these polymer thin films could be fabricated using low temperature process based on wet process owing to good solubility in volatile organic solvents. Therefore, 4-n-alkyloxyphenoxymethyl-substituted polystyrenes are a potential candidate for LC orientation layers with next generation applications with low temperature wet processes.
In view of these achievements, we believe that our work represents a timely methodological advance and breakthrough in the field of polymer science and thus is appropriate for a journal with the scope and wider readership of “Polymers”.
I hereby certify that this manuscript consists of original, unpublished work which is not under consideration for publication elsewhere.
We are excited to share our manuscript with you and look forward to hearing good news from you soon.
Thank you very much for your time and consideration for the process.
Sincerely (on behalf of all authors),
Prof. Hyo Kang
Associate Professor
Department of Chemical Engineering
Dong-A University
Busan 49315, Republic of Korea
Tel: +82 51 200 7720
Fax: +82 51 200 7728
E-mail [email protected]
